# Infrared Thermography Sensor for Disease Activity Detection in Rheumatoid Arthritis Patients

**DOI:** 10.3390/s19163444

**Published:** 2019-08-07

**Authors:** Jolanta Pauk, Agnieszka Wasilewska, Mikhail Ihnatouski

**Affiliations:** 1Mechanical Engineering Department, Automatics and Robotics Faculty, Bialystok University of Technology, Wiejska 45C, 15-351 Bialystok, Poland; 2Scientific and Research Department, Yanka Kupala State University of Grodno, Elizy Azeska 22, 230023 Grodno, Belarus

**Keywords:** thermography, fingers, rheumatoid arthritis, image processing, disease activity level

## Abstract

A recent review of thermography studies in rheumatoid arthritis shows limited data about disease activity and mostly focuses on differences between the thermography of rheumatoid arthritis patients and typical subjects. A retrospective study compared patients with high disease activity (n = 50), moderate disease activity (n = 16), and healthy participants (n = 42), taking into account demographic, clinical, laboratory, and thermography parameters. We applied an infrared thermography sensor and a fingers examination protocol. Outcomes included the mean temperature of five fingers of a hand: In static, post-cooling, post-rewarming, the total change in mean temperature of fingers due to cold provocation, the total change in mean temperature of fingers due to rewarming, the area under the cooling curve, the area under the heating curve, the difference between the area under the rewarming and the cooling curve, and temperature intensity distribution maps. For patients with high disease activity, a lower area under the heating curve and a lower difference between the area under the rewarming curve and the cooling curve were observed, as well as a smaller total change in mean temperature due to rewarming, compared to patients with moderate disease activity (*p* < 0.05). Our study findings could be helpful in patients with an equivocal clinical examination.

## 1. Introduction

Rheumatoid arthritis (RA) is the most common inflammatory and systemic connective tissue disease with an autoimmune background, characterized by symmetrical arthritis, non-articular changes, and systemic complications [1,2,3]. The annual incidence of rheumatoid arthritis (RA) has been reported to be around 40 per 100,000. In the European population, the prevalence of RA is estimated at 0.5–1.1%, and in the United States it ranges from 0.53–0.55% [4,5,6]. RA can occur at any age. However, the highest incidence is observed in people over 40 years old, three times more often in women than in men. RA can be mild, moderate, or severe, and patients can experience pain, heat, and swelling in the disease process [7,8]. The disease is characterized by disorders related to innate immunity in the processes of complement activation by the antigen-antibody complex as well as adaptive immunity, which manifests itself in the immune response against its antigens. Among the antibodies attacking elements of the body’s tissues, the best known are autoantibodies directed against the fragment crystallizable region (Fc region - the region of an antibody that interacts with cell surface receptors (Fc)) of class G immunoglobulins (IgG), i.e., rheumatoid factor (RF). Patients with RA who have high titres are at increased risk of developing non-articular disease symptoms, such as rheumatoid nodules or vasculitis or disorders of the nervous system, digestive system, skin, liver, eyes, or pleural [9]. A natural process that occurs as a result of inflammation is the modification of proteins by their citrullination. A characteristic feature of RA is the occurrence of an adaptive immune response against these proteins, which is manifested by the presence of so-called anti-cyclic citrullinated peptide antibodies (anti-CCP). Anti-CCP is present in the patient’s blood many years earlier than the appearance of clinical features of the disease.

Currently, the diagnosis of RA is based on criteria of the American College of Rheumatology (ACR) [3], which includes clinical, biological, and radiological findings. The disease most frequently affects the wrist, metacarpophalangeal and proximal interphalangeal joints [8]. Computed tomography (CR) is a useful tool in detecting structural abnormalities of bones and joint space narrowing [10]. However, this technique is not able to detect soft tissue synovitis and the earliest stages of bone erosion, thus reducing information relevant for the rheumatologist about the activity of the disease [11,12]. Additionally, magnetic resonance imaging (MRI) provides visualization of all the tissue components involved in the RA and sufficiently detects bone erosions at an early stage of the RA disease. However, it is difficult to use this method for patients’ screening [13,14]. Another technique is the ultrasound (US), which enables detecting early bone erosions and soft tissue abnormalities. A power Doppler examination can also be used, but it is less sensitive than radiography for detecting bone erosions [15]. Van den Berg et al. [16] proposed a portable ultrasound and photoacoustic imaging (PAI) system with a hand-held probe to detect clinically evident synovitis. The advantage of this technique is the detection of small blood vessels in the finger arthritis joint capsules, which may be hyperactive in the disease. There are strengths and limitations to each modality [17,18]; hence thermography may be a suitable sensor for joint inflammation detection in human and animals [19,20,21] Static and dynamic protocols are the two most used. The first protocol measures steady-state conditions. The dynamic includes cooling and rewarming, then captures a series of thermograms during a time, providing useful tissue information [22,23,24].

The authors of previous related research used IR thermography in medical applications for fever screening of potentially infected patients in airports [25], breast cancer detection [26,27], diabetes neuropathy [28], peripheral vascular disorders detection [29], and in psoriasis arthritis [30]. In recent studies, the role of thermography examination of hand joints for diagnosis and prediction of disease progression was evaluated [31]. Frize et al. [20] and Snekhalatha et al. [23] performed a passive thermographic examination of the hands, wrists, and knee joints of patients with RA and healthy participants. Borojevic et al. [24] collected the thermographic images of hands for healthy subjects, patients with rheumatoid arthritis, and patients with osteoarthritis [24] and found, that that heat distribution over the skin surface differs between the patients with rheumatoid arthritis and osteoarthritis. Other researchers analysed the use of active thermography in the diagnosis of RA. Rusch et al. [19] observed a slower flow of blood from pathological venous vessels in RA patients during the heating process. Leijon–Sundqvist et al. [32] examined the temperature of palmar and dorsal parts of the hand. This approach explained tissue reperfusion represented by the surface area of particular heating fingers. In [33], authors determined that RA patients without active inflammation of the hands demonstrate a significantly higher mean temperature compared to healthy individuals. The main challenge in RA evaluation is an assessment of RA activity levels in case of no univocal clinical features. This study purpose was to demonstrate the use of infrared thermography sensor to assessment of disease activity in patients with rheumatoid arthritis and compare it to a group of healthy age-matched subjects.

## 2. Materials and Methods

### 2.1. Subjects

This is a cross-sectional study which evaluates rheumatoid arthritis patients who were referred to the Clinical Hospital of Bialystok Medical University at Rheumatology and Internal Diseases Clinic from January 2017 to December 2018. The total number of RA patients studied before exclusion criteria was 155. Exclusion criteria for patients included respiratory diseases, cardiovascular diseases, dermatological diseases, rheumatic diseases other than RA, and treatment other than biological (Figure 1).

Finally, we recorded a set of demographic, clinical, and laboratory measurements and thermography for 66 patients with a mean age of 53.8 years in two groups: High disease activity (n = 50) and moderate disease activity (n = 16), based on the disease activity score 28 (DAS28). Compared to an initial value, disease activity in the patient was classified as follows: Moderate activity 5.1 > DAS28 > 3.2 and high activity DAS28 > 5.1 [3]. A total of 42 healthy subjects were involved in this study as a control group. The patients participated in the study with their consent, according to the declaration of Helsinki. The Polish Regional Committees have approved this study for Medical and Health Research Ethics (Medical University of Bialystok, No. R-I-002/16/2016).

### 2.2. Clinical Disease Assessment and Treatment Details

At the time of the thermography examination, we collected information about the disease duration (in months), clinical joint examination findings, including tender joint count (TJC), swollen joint count (SJC), and DAS28. Immunological factors included: Rheumatoid factors (RFs), anti-citrullinated peptide antibodies (ACPA), concomitant treatment, and acute phase reactants: C-reactive protein CRP, erythrocyte sedimentation rate (ESR), and blood cell counts: Erythrocytes, leukocytes, and thrombocytes. For each patient, a detailed record was compiled of their medication at the time of the thermography examination. Most patients were treated with non-steroid anti-inflammatory drugs, 76% of the patients treated with glucocorticoids. Meanwhile, methotrexate has also been used by 76% of patients.

### 2.3. Infrared Camera as a Sensor for the Level of Rheumatoid Arthritis Detection

Thermography examination was performed using a FLIR E60bx (Wilsonville, OR, USA) thermal imaging camera with a resolution of 320 × 240 pixels (Figure 2). The camera sensitivity was <0.405 °C, the accuracy was <2%, and the factory calibration spectral sensitivity ranging was 8–12 µm. It had a geometric resolution of 1.5 mrad and, a 30–20° field-of-view lens with a minimum focus distance of approximately 20 cm and the thermal resolution was 0.03 K. The accuracy of the absolute temperature measurement was <2 K. For all analysed regions, the emissivity was ε = 0.98. The camera registered an object temperature range from −20 °C to +120 °C. The registration process was carried out with an imaging frequency of 60 Hz. The settings used were: Patient acclimatisation 15 min; camera distance 1 m from the fingers; air humidity 55%; emissivity 0.98; air and ambient temperature 23 °C. The thermograms were acquired three times from fingers in sitting position into three stages: (1) The temperature in static; (2) the temperature video recorded during cooling; (3) the temperature video recorded during rewarming.

In order to determine the required time for fingers cooling and rewarming, a series of periodic measurements were performed. The main criteria were decreasing the finger temperature by 6 °C during cooling and temperature stabilisation during rewarming. The fingers were cooled in water at 0 °C ± 0.2 °C for 5 s, and the time of rewarming was 180 s. The patients were routinely asked not to drink alcohol, coffee, or caffeinated drinks for 24 h, not to smoke for 2 h, and not to do physical activity 1 day before measurement. The examinations were done between 13:00 and 14:00, according to the Glamorgan protocol [34]. The thermography examination included fingers of both hands from the dorsal plane comprising the thumb, index finger, middle finger, ring finger, and little finger.

### 2.4. Image Processing

Thermographic images were analysed, and the temperature values of the fingers were extracted using the Matlab software (MathWorks, Natick, MA, USA). Dynamic data were obtained from three video frames: Before cooling, post-cooling, and post-rewarming. The proposed methodology to diagnose RA disease activity levels included a few steps. Firstly, we proposed a protocol for the acquisition of reliable thermographic data. Secondly, the processing of the images acquired by the thermovision sensor included pre-processing techniques, such as conversion of a thermal image into a greyscale image and improving the grey image in order to obtain a clear border between the object and the background. The average filter was used for the image smoothing, and the noise reduction was performed by using a median filter. To improve the image, normalization using gamma correction was applied. Those three procedures can be applied separately or together in any combination. The average and the median filters were performed using a 3 × 3 mask. The Gamma correction was verified by controlling the maximum on the image brightness histogram. The increase of the peak value and a decrease of the peak’s width indicated that a uniform background was obtained. These three procedures were performed in order to more easily separate an object from the background. The procedure of separating an object from the background (image thresholding) consisted of two steps. In the first step, a grey scale image was converted into a binary image. It was done in two ways. Firstly, the least minimum on the histogram lying between two local maxima was found, and the balanced histogram thresholding (BHT) was applied [35]. This method is always used in finding image thresholding for the first or the second frame if the object has a clear border with a background. Secondly, the image threshold between the second and third frame checks if the object does not have a clear border with the background. During the second step, the noise was filtered between the object and the background, and the filtration consisted of the iterative, using morphological operations (dilatation and erosion). The result of morphological operations was monitored at each iteration. If the effect was not minimized within 10 iterations, then the algorithm returned to the first step. The procedure of image segmentation also consisted of two steps. The selecting of the middle line of the fingers is not trivial since the fingers are not the axisymmetric figures. Firstly, the extraction of the fingers by using skeletonization was performed using an iterative algorithm that removed the outer layers of the hand image. The iterative algorithm moved the 3 × 3 mask and matched it with the parts of an image. Secondly, object identification was performed by using a modified depth first search (DFS) [36]. The five profiles of the recognized fingers with one-pixel width were a region of interest (ROI), Figure 3.

The thermographic image superimposed on the segmented image was then analysed. We recorded information about the mean temperature of the ROI: In static (*T_init_*); post-cooling (*T_C_*); post-rewarming (*T_R_*). Data at cooling removal and over 180 s generated rewarming curves. Then we counted the total change in mean temperature of the ROI due to cold provocation *(*∆*T_C_*), the total change in mean temperature of the ROI due to rewarming *(*∆*T_R_*), the area under the cooling curve (*Sc*), the area under the heating curve *(S_R_*), the difference between the area under the rewarming curve and the cooling curve (*S_R_–Sc*), and the temperature intensity distribution maps.

### 2.5. Statistical Analysis

Method reproducibility was the coefficient of variation (%CV = σ/μ × 100). Data were stratified into three groups: High disease activity, moderate disease activity, and healthy subjects The values were presented as means and standard deviations. To verify the hypothesis of a normal distribution of the analyzed variables, the Shapiro–Wilk test was used. To further analyse variables with normal distribution, a parametric test (Student’s test) and other nonparametric tests were used. To determine the significance of differences between the three groups, the Mann–Whitney test was used. Finally, correlations between clinical and thermography data were assessed using the Spearman rank test. A *p*-value of <0.05 was considered a statistically significant result. The statistical tests were performed with the use of Statistica 13.1 (StatSoft, Krakow, Poland).

## 3. Results

### 3.1. Patients Demographic

The RA patients were predominantly female (83.3%). We compared the three groups of interest, comprising 50 patients with high disease activity, 16 patients with moderate disease activity, and 42 healthy participants. The baseline characteristic for RA groups and healthy participants are presented in Table 1.

Clinical parameters (CRP and ESR, *p* < 0.05) significantly differentiated the typical subjects and RA patients. Additionally, the discriminating parameter for the high disease activity group and moderate disease activity group was the mean duration of the disease (*p* < 0.05).

### 3.2. Thermograms Analysis

Using the methodology described in Section 2, three videos per patient were taken for fingers from the dorsal side (Coefficient of Variation (CV) was 0.04%). A comparison between right and left fingers showed no significant difference in temperatures (*p* > 0.05). The example of dynamic thermal imaging outcomes registered for a randomly selected subject from the experimental group, high disease activity (HD), moderate disease activity (MD), and healthy (H), was presented in Figure 4.

An average value of thermal imaging outcomes for high and moderate disease activity compared to healthy subjects are presented in Figure 5.

During the static measurement, the average temperature of the fingers for RA patients with moderate and high disease activity was similar (31.3–32.3 °C for high disease activity group vs. 31.5–32.3 °C for moderate disease activity group, *p* > 0.05). However, post-cooling (*T_C_*), the tissues of the fingers of patients with moderate disease activity cooled more strongly than patients with high RA activity. The thermal response of tissues during the heating of the fingers (*T_R_*) differentiated the patients to a greater extent (*p* < 0.05). For patients with high disease activity, the impaired vascular flow and characteristic features of ischaemia of the fingers skin were observed, which was manifested by much slower heating of these areas of the hand. These features have not been found among people with moderate disease activity (*p* > 0.05). Statistically significant differences between all groups were observed for the total change in mean temperature due to rewarming *ΔT_R_*, *p* < 0.05. The temperature post-cooling and post-rewarming were significantly higher in healthy participants compared to the high disease activity group (*p* < 0.05). Additionally, the change in mean temperature due to rewarming was smaller, and the difference between the area under the rewarming curve and the cooling curve (*S_R_–S_C_*) was lower for the high and moderate disease activity groups compared to the healthy group, *p* < 0.05. Dynamic outcomes were significantly different among patients with high disease activity and the group with moderate disease activity. For patients with high RA activity, a lower area under the heating curve *S_R_* and a lower difference between the area under the rewarming curve and the cooling curve (*S_R_–S_C_*) value was observed, as well as a smaller total change in mean temperature due to rewarming compared to patients with moderate disease activity (*p* < 0.05). Figure 6 illustrates the colour image, corresponding static IR image, and dynamic IR images over 185 s for a healthy subject and a randomly selected patient with high and moderate disease activity. Colours indicate the thermal intensity in (a) static; (b) after 5 s of cooling, and (c) after 180 s of rewarming.

Before cooling (static, time 0), the average temperature of the fingers was lower for the patient with high disease activity vs. the patient with moderate disease activity. For the patient with a DAS28 = 4.2, the temperature post-cooling (*T_C_*) of the all fingers decreased by an average of 6.2 °C, while, for the patient with DAS28 = 6.9, the fingers cooled by an average of 4.6 °C. The increase in the average temperature of the fingers post-rewarming (*T_R_*) for the patient with DAS28 = 4.2 was 6.2 °C, and the area under the heating curve *S_R_* was in the range of the healthy participants (764.4 °C∙s). Meanwhile, for the patient with DAS28 = 6.9, the temperature post-rewarming of fingers increased by an average of 1.1 °C and the area under the heating curve *S_R_* was significantly lower (186.3 °C∙s). The patient with high disease activity during the thermal recovery phase exhibited very poor finger warming. It was found that impaired vascular flow was strongly associated with active inflammation, swelling, and pain in the joints.

### 3.3. Relationship between Thermography and Clinical Data

For patients with moderate disease activity, a negative and statistically significant correlation was found between the total change in the mean temperature due to rewarming and the DAS28 for all fingers (R = −0.97, *p* < 0.05). The negative correlation can be explained by the fact that higher disease activity is associated with the process of cartilage and bone tissue destruction, which manifestds itself in the reduced temperature of the affected area. Additionally, the number of tender joints correlated significantly with the total change in mean temperature due to rewarming (R = −0.96, *p* < 0.05). Furthermore, the number of tender joints correlated with the mean temperature post-rewarming (R = −0.92, *p* < 0.05). For patients with high disease activity, statistically significant correlations were found between the temperatures of all fingers post-cooling and the RF (R = 0.72, *p* < 0.05) and anti-CCP (R = 0.75, *p* < 0.05). The presence of anti-CCP in the patient’s serum is a predisposing factor to the inflammation of the blood vessels, which manifests itself as ischemia of the fingers and occurs in a disease lasting at least 10 years.

## 4. Discussion

Although there are many tools for diagnosis of the state of inflammation in rheumatoid arthritis, it is essential to know if infrared thermography can be used as a supportive diagnostic tool to differentiate the level of inflammation as well. The aim of this study was to examine dynamic IR thermography for the detection of the RA disease activity status. The examined group of patients was characterized by high disease activity with the value of DAS 28 > 5.1, moderate disease activity with 5.1 > DAS28 > 3.2, and healthy subjects with DAS28 < 2.6. Two main challenges in the disease activity level detection using IR sensors were recognized in this research. First, we proposed tools for image processing algorithms, determining the region of interest (ROI). Second, we established a target time for cooling and rewarming in the aim of differentiating the disease activity level. Concerning thermographic studies in RA, different image processing techniques have been used. The earliest of them involved the manual determination of the area of interest, while the most recent introduced elements of automation. In 2011, Frize and others [20] made thermograms of the hands, wrists, and knee joints in a group of healthy participants and patients with RA. Regions of interest were selected manually from grayscale images, using anatomical areas as reference points to locate the synovium. In [30], Ismail et al. selected 14 regions of interest located on the hand’s dorsum, corresponding to the interphalangeal joints, both proximal and distal, metacarpophalangeal joints, nails, and inter-bone muscles. The number of regions of interest may be different; however, due to the risk associated with the omission of inflamed tissue, an automatic method of identifying lesions in RA should be applied. The latest study on RA patients described infrared thermography [23] as the sensor of the temperature distribution of the palm of patients with RA and healthy patients, with the use of image segmentation algorithms based on the k-means method to quantify disease changes, where ROI was determined automatically. However, to our knowledge, no previous studies provided a clear cut-off between the inflammatory changes that can differentiate RA values on the disease activity level using the IR sensor.

The time for cooling and rewarming was 5 and 180 s, respectively, determined during the experimental procedure. The temperature post-cooling (*T_C_*) compared to the static measurement decreased and was between 6.2–8.3 °C for the patients with high disease activity, 6.3–8.9 °C for patients with moderate RA activity, and 6.1–8.1 °C for healthy subjects. Our study shows that the reperfusion processes occurring during the thermal provocation in the form of a cooling stimulus occur differently in patients with RA and healthy patients. The response to cooling between patients with varying RA severity is also diversified. It was observed that, in the moderate disease activity group, swelling of the synovium was manifested by elevated skin temperature around the joints, with occasional extra-articular symptoms, including surrounding tissues. In high disease activity with a severe course, extra-articular symptoms, from small and medium blood vessels that supply blood to the skin, nerves, and deeper structures (in the finger), were common [31,37]. Our findings show that IR imaging detects differences in the total change in mean temperature due to rewarming. ∆*T_R_* increased: 3.4–3.9 °C for patients with moderate disease activity vs. 1.8–2.6 °C for patients with high disease activity. We also found that *S_R_* is a useful parameter, which differentiates the status of inflammation. For high and moderate disease activity, its range of value was 316.7–422.3 °C∙s and 511.6–580.9 °C∙s, respectively, whereas the range of value for healthy subjects was 688.1–731.9 °C∙s. The higher amount of *S_R_* for moderate disease activity compared to high disease activity implied increased thermal activity, e.g., greater perfusion, metabolism, and rapid endothelial cell proliferation. Significant changes in ROI thermal activity were also found in contrast (*S_R_*–*S_C_*) with the lowest mean value (298.3 °C∙s) for high disease activity and the highest value for the healthy group (716.2 °C∙s). The results suggest higher thermal activity of the uninvolved tissue, likely due to more perfusion and metabolism, resulting in greater susceptibility to cold exposure and greater capacity to recover than RA patients with high disease activity. Overall, in patients with high DAS28 > 5.1, impaired reperfusion processes were observed in the fingers, and analysis of the dynamics of temperature changes post-cooling showed disturbances of tissue reperfusion, which may be evidence of ischemic processes resulting in an abnormal vascular flow in the cutaneous blood vessels of the fingers (Figure 6).

Some studies show a significant correlation between thermographic findings and disease activity in rheumatoid arthritis. Devereaux et al. [38] presented a significant correlation (*p* < 0.001) between articular index (AI) thermography, partial AI, Mally’s scale, compression force (GS), morning stiffness (MS), ESR, and the pain scale (PS). CRP showed a higher correlation between both joint indexes (*p* < 0.05) and the Mally scale (*p* < 0.001) than with thermography. Significant correlations of AI, the Mally scale, GS, morning stiffness, erythrocyte sedimentation rate and PS with thermography confirm the validity of their use in combination with this imaging method for assessing rheumatoid arthritis. In the present study, it was proved that the use of cold provocation revealed the occurrence of statistically significant, high correlations that did not occur for static measurement (before cooling). In the group with moderate disease activity, correlations were observed between the total change in mean temperature due to rewarming and DAS28 and also the number of tender joints (*p* < 0.05). In the group with high disease activity, temperatures measured immediately post-cooling significantly correlated with RF and anti-CCP (*p* < 0.05). The serological parameters in the patient’s serum predispose to a more severe course of the disease and extra-articular symptoms, such as vasculitis (inflammation of the blood vessels) [39,40]. In the present study it manifested as the ischemia of the fingers.

The shortcoming of this paper is the small number of retrospectively evaluated patients in the moderate disease activity group. For this group, the interpretation of the data must be treated as preliminary. More studies and patients are necessary to get more reliable information. Further studies are required to achieve confirmation of these findings and provide results for patients with low-level disease activity.

## 5. Conclusions

This study shows that dynamic infrared thermography detects the RA disease activity level and can be used in clinical practice as a supportive tool in diagnosis from several reasons: It is a fast and straightforward method, safe, flexible, and budget-friendly. It has some limitations that should be taken into account when applied in medical practice. Infrared thermography depends on the sensor and the experimental setup. The sensor presented in this study has a sufficient thermal resolution. This was proved during adequate setup and testing procedures involving two phases: Finger cooling and finger rewarming. IR thermography is widely used in different medical applications. Recent advances in this field allow this technology to detect breast cancer, diabetes, and neuropathy etc. Within this study, it was demonstrated that IR thermography differentiates the disease activity level and shows characteristics of inflammation in patients with RA, which could be helpful in diagnosis, and treatment of patients with an equivocal clinical examination. Future works should be concentrated on using different sensors with improved sensibility and presenting new image processing techniques to enhance ROI detection from infrared thermal images.

## Figures and Tables

**Figure 1 sensors-19-03444-f001:**
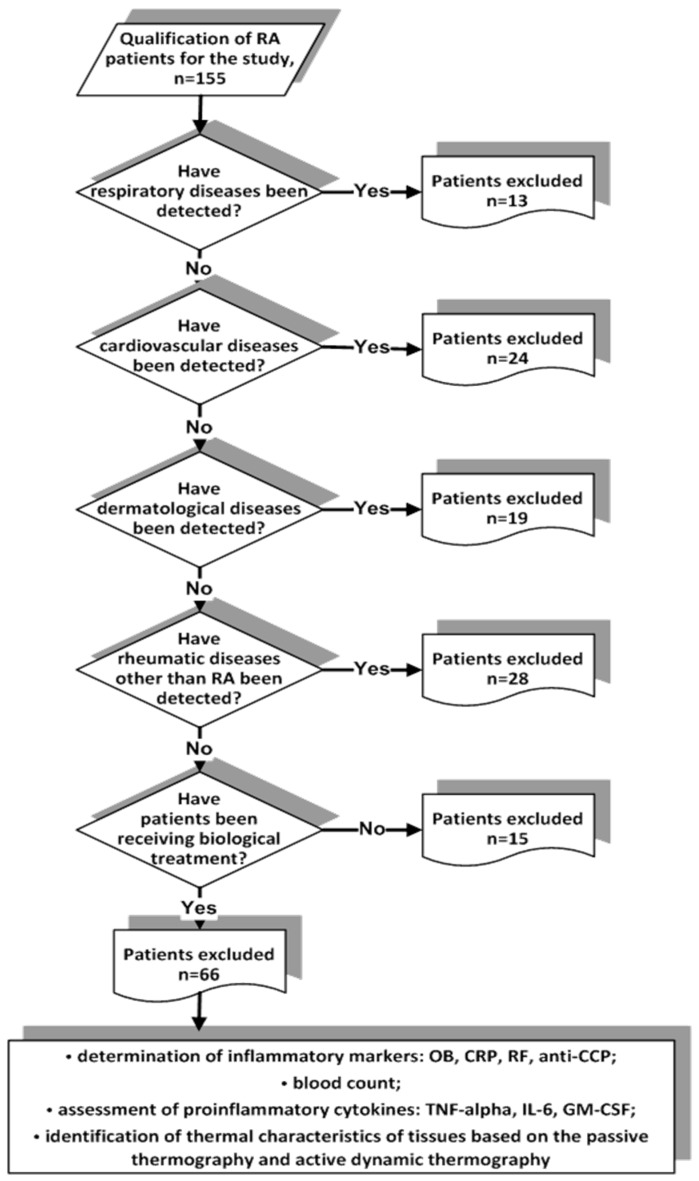
Flowchart for patients classification and data collecting.

**Figure 2 sensors-19-03444-f002:**
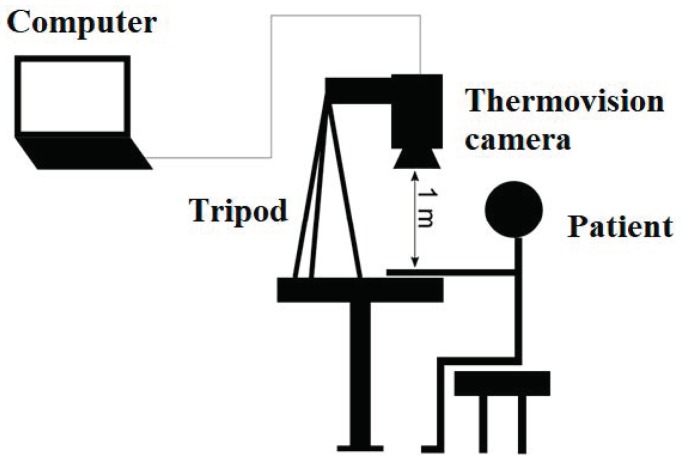
Set up for measurement.

**Figure 3 sensors-19-03444-f003:**
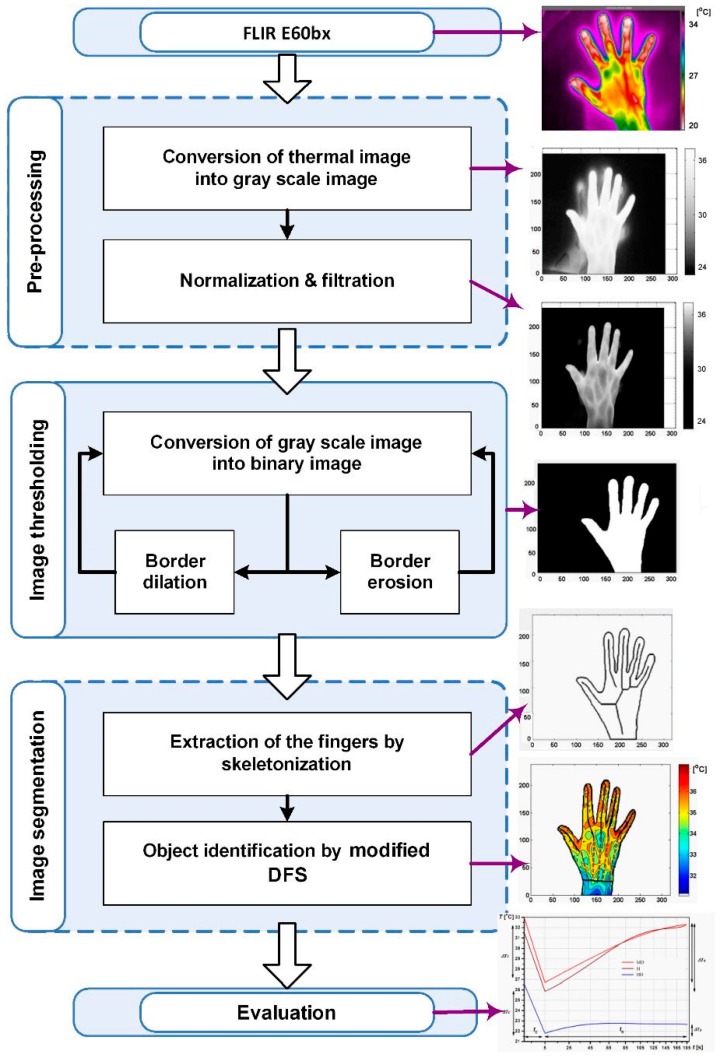
Workflow of thermographic image processing.

**Figure 4 sensors-19-03444-f004:**
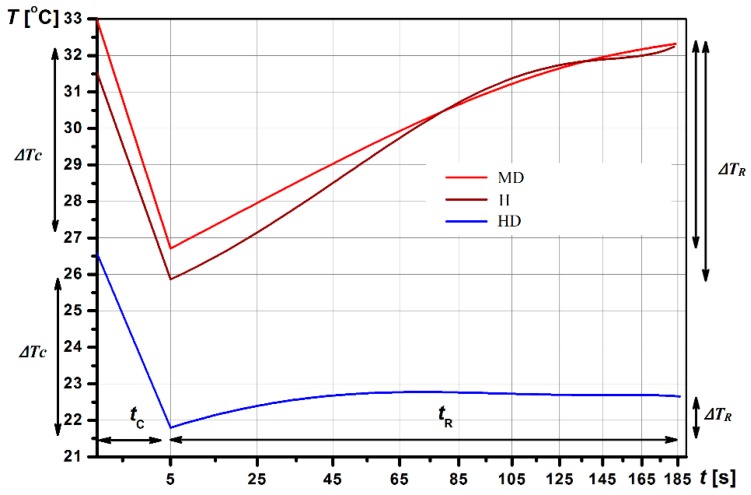
Dynamic thermal imaging outcomes registered for a randomly selected subject from the experimental group: High disease activity (HD), moderate disease activity (MD), healthy (H), *t_c_ =* cooling, *t_R_ =* rewarming, ∆*T_R_ =* total change in mean temperature due to rewarming, ∆*T_C_* = total change in mean temperature due to cold provocation,

**Figure 5 sensors-19-03444-f005:**
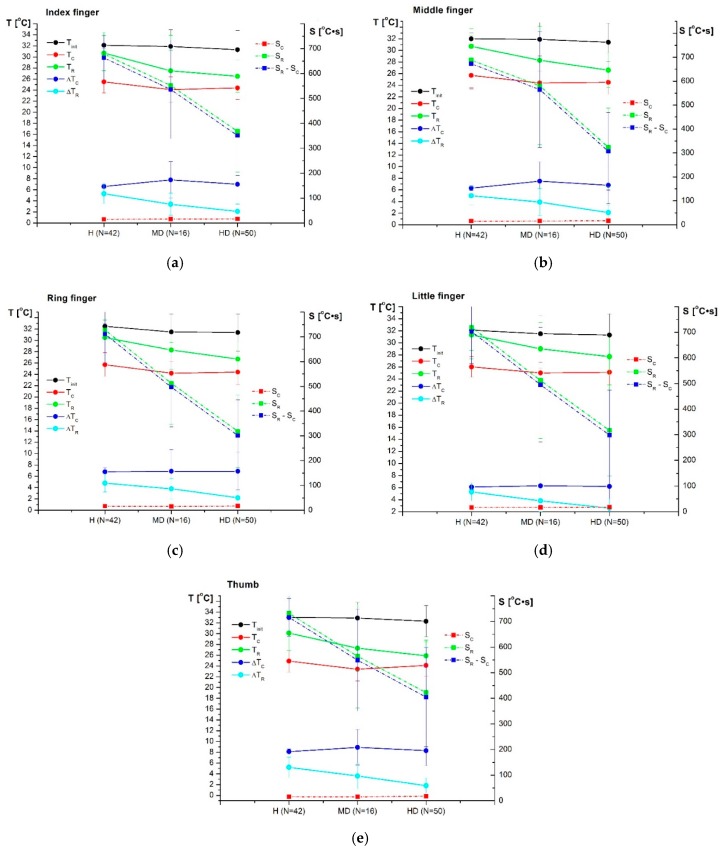
Mean (SD) value of thermal imaging outcomes for high and moderate disease activity compared to healthy subjects: (**a**) Index finger; (**b**) middle finger; (**c**) ring finger; (**d**) little finger; (**e**) thumb.

**Figure 6 sensors-19-03444-f006:**
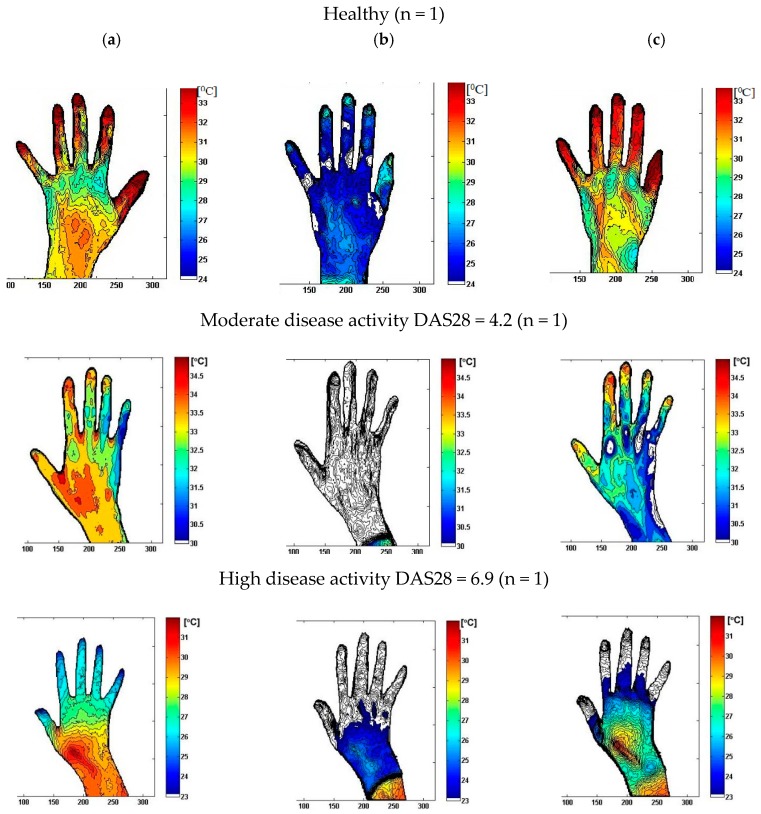
Distribution maps collected for a randomly selected subject from the experimental group: High disease activity (HD), Moderate disease activity (MD), Healthy (H): (**a**) In static; (**b**) post-cooling; (**c**) post-rewarming.

**Table 1 sensors-19-03444-t001:** Baseline characteristic of rheumatoid arthritis (RA) patients and healthy subjects (N = 108). All data are expressed as the mean (SD).

Disease Activity	High Disease Activity Group (N = 50)	Moderate Disease Activity Group (N = 16)	Healthy Group (N = 42)
Age (years)	51.9 (7.3)	55.4 (9.2)	54.1 (3.2)
Duration of the disease (years)	13.1 (2.3)	8.5 (2.8) *	0.0 (0.0)
Erythrocyte sedimentation rate ESR (min/h)	43.8 (5.7) *	39.4 (6.4) *	18.2 (6.4)
C-reactive protein CRP (mg/mL)	23.7 (7.7) *	20.0 (7.1) *	3.0 (1.3)
Erythrocytes (mln/µL)	4.4 (0.6)	4.5 (0.4)	4.4 (0.5)
Leukocytes (G/L)	7.3 (2.4)	7.2 (2.1)	5.1 (2.4)
Thrombocytes (G/L)	300.3 (38.9)	263.4 (49.4)	276.5 (37.9)
Disease activity score 28 DAS28	5.8 (0.6)	4.8 (0.3)	2.0 (0.3)
Number of tender joints	8.9 (2.8)	5.7 (2.0)	0.0 (0.0)
Number of swollen joints	5.9 (2.1)	3.7 (1.5)	0.0 (0.0)

* (*p* < 0.05).

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
