# Peer review of "Infrared Thermography Sensor for Disease Activity Detection in Rheumatoid Arthritis Patients"

_sensors, 2019, doi:10.3390/s19163444_

Round 1

Reviewer 1 Report

The authors assessed IR thermography for detecting the severity of RA patients. The proposed method can be used as a supportive diagnostic/screening tool with the advantage of safe, flexible, and budget friendly. The paper is generally well written with a very clear objective.

Suggested changes:
1. The literature references are not sufficient: The medical applications of IR thermography should be introduced more. For example, the thermography has been applied for fever screening of potential infected patients in airports (Int J Infect Dis. 2017 Feb;55:113-117. doi: 10.1016/j.ijid.2017.01.007. Epub 2017 Jan 16.), and breast cancer detection (J Epidemiol Community Health. 1990 Jun; 44(2): 112–113. doi: 10.1136/jech.44.2.112)

2. Table 2 should be better replaced by a line graph.

Author Response

The reviewer has presented us with important comments and requests, which we believe have improved the quality of our manuscript significantly. In the following pages, we address each and every comment in detail. Each comment is marked by the reviewer’s original text indexed for convenience of reference, and explains what and how changes were implemented in the new submission. We hope the revised manuscript is much closer to the high standard of Sensors, and we should be very glad to address any further issues in subsequent revisions.

The literature references are not sufficient: The medical applications of IR thermography should be introduced more. For example, the thermography has been applied for fever screening of potential infected patients in airports (Int J Infect Dis. 2017 Feb;55:113-117. doi: 10.1016/j.ijid.2017.01.007. Epub 2017 Jan 16.), and breast cancer detection (J Epidemiol Community Health. 1990 Jun; 44(2): 112–113. doi: 10.1136/jech.44.2.112)

Answer: We cited five references, additionally (from 25 to 29 position in the reference list).

2. Table 2 should be better replaced by a line graph.

Answer: Done.

All changes, also grammatical, we marked in red colours.

Reviewer 2 Report

The manuscript is nice to read, and I agree with the presentation. I only noticed three terminology-items that the authors must correct. These are:

Line 51 : " ..disorders of the nervous system, digestive system, .." So: add system (2 x)

Line 92 : " ...the palm." Must be corrected into : "the hand." - simply because there is no dorsal side of the palm (= nonsense).

Line 204 : the abbreviation "ROI" must be explained as "region of interest" from the first time that it is mentioned. Now we must wait until lines 343-344, which is uncomfortable.

Two small improvements may give it some extra boost.

First is the anatomical use of the words "internal organs" in line 373. This causes some confusion, because currently we mean by this: heart, lungs, liver, intestines, etc. Obviously, that is not what the authors intend, I presume. So the authors should replace it by the words "deeper structures (in the finger)", so that their meaning is clear.

Second the authors should add in their literature review, from about line 70, this novel technique by Van den Berg et al. from 2017, see second attachment. It means a real update, although this finding is not yet used as a standard now. Still their photoacoustics technique has the advantages of imaging quite small blood vessels, in the finger arthritis joint capsules, that may be hyperactive in disease, so these are directly related to the thermography method that the authors have applied. Reumatologists like this very much.

Author Response

The reviewer has presented us with important comments and requests, which we believe have improved the quality of our manuscript significantly. In the following pages, we address each and every comment in detail. Each comment is marked by the reviewer’s original text indexed for convenience of reference, and explains what and how changes were implemented in the new submission. We hope the revised manuscript is much closer to the high standard of Sensors, and we should be very glad to address any further issues in subsequent revisions.

Line 51 : " ..disorders of the nervous system, digestive system, .." So: add system (2 x)

Answer: Done.

Line 92 : " ...the palm." Must be corrected into : "the hand." - simply because there is no dorsal side of the palm (= nonsense).

Answer: Done.

Line 204 : the abbreviation "ROI" must be explained as "region of interest" from the first time that it is mentioned. Now we must wait until lines 343-344, which is uncomfortable.

Answer: Done.

First is the anatomical use of the words "internal organs" in line 373. This causes some confusion, because currently we mean by this: heart, lungs, liver, intestines, etc. Obviously, that is not what the authors intend, I presume. So the authors should replace it by the words "deeper structures (in the finger)", so that their meaning is clear.

Answer: Done.

Second the authors should add in their literature review, from about line 70, this novel technique by Van den Berg et al. from 2017, see second attachment. It means a real update, although this finding is not yet used as a standard now. Still their photoacoustics technique has the advantages of imaging quite small blood vessels, in the finger arthritis joint capsules, that may be hyperactive in disease, so these are directly related to the thermography method that the authors have applied. Reumatologists like this very much.

Answer: Done. The cited reference is under position [16] in the reference list.

All changes, also grammatical, we marked in red colour.